# Development of Marina Services in the Context of Sustainable Water and Coastal Tourism

Aelita Skaržauskienė [1] , Daiva Labanauskaitė [2] and Ana Timonina-Mickevičienė [2,*]

1   Department of Entertainment Industries, Vilnius Gediminas Technical University, Saulėtekio al. 11, LT-10223 Vilnius, Lithuania

2   Faculty of Social Sciences and Humanities, Klaipeda University, Herkaus Manto str. 84, 92294 Klaipėda, Lithuania

*   Correspondence: ana.timonina-mickeviciene@ku.lt

**Abstract:** The popularity of water and coastal tourism is steadily increasing. Marinas, an essential part of water tourism activity, are complex organizations with heterogeneous business structures with numerous suppliers of various tourist services. The purpose of this research was to determine the components of marina services that are relevant for visitors of the marina in the context of sustainable water and coastal tourism. The study's preliminary results show that the orientation towards sustainable tourism significantly influences the behavior of visitors. Consumers of marina services would like the marinas to offer more services: not only boatyard facilities, but also catering, accommodation services, safe infrastructure for children, additional entertainment, and public events.

**Keywords:** marina services; sustainable tourism; water tourism; marine tourism





## 1. Introduction

The popularity of water and coastal tourism is steadily increasing. Marinas, a vital part of water tourism activity, are complex organizations including heterogeneous business structures with numerous suppliers of various tourist services. Their purpose is to ensure the provision of those amenities, which are collectively referred to as 'marina services.' Some authors define [Paker N., Orams M., Lueck M.] a marina as a specially designed harbor with moorings for pleasure yachts and small boats. From a tourism development perspective, marinas can be characterized as coastal destinations and analyzed through the prism of place marketing [1,2]. When evaluating marinas, it is essential to take into consideration what marina visitors seek [3]. Marinas are the most advanced form of specialized tourist ports, and are expressly built for nautical tourism in well-protected areas. Along with berths and ancillary technical services, these marinas offer a wide range of accommodation, dining, shopping, entertainment, and leisure facilities to provide a 'resort atmosphere' [4].

Even though marinas contribute to the diversification of tourism flows, developing niche tourism is crucial to creating a unique selling proposition oriented toward sustainable tourism. Strengthening the coherence of tourist demands and residents' recreational interests is a helpful measure for marinas to stand out in a competitive environment [5]. Two forces drive the transition to sustainable tourism and the practical application of its principles: the goals of the 2030 Agenda for Sustainable Development (2018) [6] and the search for the optimal balance of tourism at national and local levels. The latter includes the involvement of cities, protected areas, businesses, and communities, as well as the growing awareness of visitors. The scientific sources [Bevenolo C., Spinelli C., Kovačic M., Silveira L., Tosun N.] on marina development in different countries focus on the overall growth of water tourism and the development of tourism-related infrastructure. However, some issues remain overlooked. Firstly, the expectations of the target audiences regarding facilities of marina services are not analyzed comprehensively in scientific literature. Secondly, the

compatibility of expected service components with the principles of sustainable tourism development needs more attention from a scientific point of view.

This research study aims to identify the main components of marina services that are relevant for different visitor groups in sustainable water and coastal tourism. The paper consists of four central parts. The first part relates to the theoretical research about the specifics of marina services and their focus on visitors' needs. The next part reveals actual trends in relevant services transitioning to sustainable tourism. The empirical part covers the insights on how the behavior of marina visitors in the Klaipeda region shifted towards sustainable tourism. The implemented quantitative survey identified the essential factors that increased the attractiveness of marina services to consumers in the South Coast Baltic Klaipeda region.

## 2. Marina Services as an Essential Part of the Maritime Tourism Industry

In the research literature, marina tourism and nautical tourism are defined variously. Nautical tourism is an aspect of tourism in general, from which it has developed as a subtype. The question of defining nautical tourism should thus be considered in the context of a general definition of tourism. Nautical tourism covers two aspects: the tourist aspect, which is the economic dominant, and the marine aspect, which refers to navigation not only by sea, but also by rivers, lakes, and channels [7].

"Marine tourism" and "nautical tourism" terms share conceptual similarities, and although there are features related to all of the terms, "nautical tourism" is considered a broader term that includes lakes, rivers, and other aquatic environments where tourists can enjoy boating activities. Furthermore, some authors state that nautical tourism has not yet developed in rivers and lakes [8] and that recreational boating and its related tourist products are limited to the coastal sea area. Once reviewed, the contents of articles related to river tourism have not been taken into account for this analysis.

Marine tourism emerged as an indispensable issue in researching oceans and coastal areas. M. Orams [9] defines marine tourism as "those recreational activities that involve travelling away from one's place of residence and whose focus is the marine environment" and refers to the latter as "those waters that are saline and affected by the tide". This definition also lists the related activities: scuba diving, snorkeling, windsurfing, fishing, watching sea mammals and seabirds, the cruise and ferry industry, all beach activities, sea kayaking, visits to coastal villages and fishing lighthouses, Maritime museums, sailing and motor boating, maritime events, and Arctic and Antarctic tourism.

The concept of maritime tourism includes all tourism activities related to the sea and the coasts and is one of the fastest-growing segments of maritime activity, which has been rapidly developing worldwide [10]. This is a specific form of tourism that includes holidays, recreation, and leisure [11]. Marine tourism involves sailing, using recreational vessels, boats, and cruise ships for leisure or business. In addition to using pleasure boats and cruises, it covers a wide range of activities such as water skiing, windsurfing, underwater fishing, scuba diving, swimming and tours of marine parks [12].

Maritime tourism grows faster compared to the primary tourism industry. There are two main reasons for this: first, particular recreational and entertainment activities; second, specific venues that are popular destinations for water tourism enthusiasts [7]. Due to its high socio-economic impact, maritime tourism offers great economic potential and opportunities to boost tourism.

A marina can be defined as "a specially designed harbor with moorings for pleasure yachts and small boats". They are complex organizations with heterogeneous business structures and several service providers who strive to ensure various recreational services, which are collectively referred to as "marina services". From this perspective, marinas can be described as a venue or a destination and analyzed from a venue marketing perspective [1].

According to the British Yacht Harbour Association (2007) [13], marinas can range from small-sized harbors capable of accommodating only a few yachts to multifunctional

harbors with a boat yard and commercial outlets capable of accommodating vessels of different sizes. IMI uses the term "Destination Resort Marina" for more closed and more complex marinas. It is defined as "a destination marina that is accessible from both water and land and combines embankments for visitors, accommodation and catering services, swimming pools, and other entertainment and recreational opportunities provided by the recreational atmosphere of the marina" [14].

Marinas can provide different tourism services, but their leading service will always be boat storage. Some visitors do not require other services, but forming other services is impossible without this essential facility. In addition to boat storage, marinas can offer boat lifting or lowering services, minor repairs, off-season storage, deck cleaning, etc. All of these are complementary products to the primary service, though the consumer expects these services to come by default with ship storage. Boat storage is unimaginable without other support services; showers and toilets for yachters, freshwater pumps, electricity, as well as refueling and repair services may be required. Other essential services include accommodation and event planning, which are specific and complex in the marina segment. The design of the basic marina service of boat storage depends on the available infrastructure. Each marina must have space for boats of different sizes. If it is a more extensive marina, pools are allocated depending on the boat size, and the berthing fees will naturally also depend on that factor.

Additional components of marina services may include accommodation and catering services, galley restocking, shopping, car parking and shuttle services, sports activities, and entertainment [15,16]. Such additional elements of marina service products help attract an increasing number of customers to marinas. Therefore, marinas can become the center of attraction in the area where they are located.

*The Development of Marina Services Focuses on the Needs of Visitors*

In a discussion about the marina product focusing on the customer needs, segmentation plays a significant role. Effective segmentation is needed because marinas offer their services to a broad range of customers, both individuals and companies. Benevolo and Spinelli, in their latest research, focus on individual customers, specifically pleasure boaters. Pleasure boaters and yachting professionals differ significantly from a marketing perspective. First, the consumption context of the marina services and success factors differ between the two groups [4].

The developments of the marina services are described by Kovačic [17] as a necessity of possibility to accommodate boats of 12–15 m or mega-yachts over 20 m long in the Mediterranean region in order to increase the service of charter voyages and additional services such as catering and entertainment in marinas. All these changes should be done because of the high demand of these services, which increases the duration of shipping, and the high public interest in marinas as a destination and complex of services.

Several studies have been conducted to determine what services are needed for marina visitors. Surveys and studies have been conducted within the scope of the South Coast Baltic European project *MARRIAGE* to determine what marina visitors were looking for and what they were paying attention to [3].

The literature and survey reviews cover numerous factors of marina service construction according to the yachters' needs and required services. Table 1 shows some factors identified during the literature review.

Table 1. Factors increasing marinas attractiveness for visitors.

| | South Coast Baltic Survey, 2017 [3] | Paker et. al., 2016 [1] | Kovačic, 2013 [17] | Cater et al. (2007) [18] | Terzi et al., 2013; [19] Tosun, 2001 [20] | Sariisik M., Turkay O., Orhan A. [21] | Results |
|---|---|---|---|---|---|---|---|
| Hospitality and attitude of employees towards consumers | x | x | | | x | | 3 |
| Developed marina infrastructure | x | | x | | x | x | 4 |
| Harbor security, safety | x | x | | | x | | 3 |
| Quality of main services, hygiene conditions | x | x | | | | x | 3 |
| Environmental friendliness/sustainability | | x | X | x | x | x | 5 |
| Additional services as catering, accommodation, entertainment | x | | x | x | x | | 4 |
| Marina as an attractive touristic place: history, nature, city | x | | | x | x | x | 4 |

A systematic review of the needs of marina users in different regions shows that visitors are mainly concerned with the quality of services provided, the friendliness and professionalism of the staff, modern marina infrastructure, and safety standards. It is essential to address these needs when developing the basic structure of marina services. Sustainability is a significant factor in marina development, as well as a demand of additional services in marina complexes.

Due to its dual structure as the interface between water and land and the diversity of services provided, successful marina management requires differentiation strategies for target groups based on consumer preferences [22]. Similarly, R. Heron and W. Juju [5] emphasize that marinas must create a "unique selling proposition" to stand out from other marinas. Competitive advantage and successful positioning of operations can be achieved by a thorough marketing analysis focused on selected market segments and developing offers for these markets based on consumer needs [5].

As the main attraction of aquatic entertainment and related services, the product of marina services is a vital part of maritime tourism [4,21,23] that can meet the needs of many users: yachters, tourists, day vacationers, athletes, or families with children. Most importantly, it should be an area with sufficient access to tourism services that can meet the needs of the majority of visitors. The supply of marina services and the development of its activities depend on different marketing trends in many cases: regional tourism marketing and positioning, partnerships between marinas and their willingness to join into clusters, groups seeking joint marketing, sharing of best practices, and other innovative marketing trends.

## 3. The Trends in Marina Activity Development in the Context of Sustainable Tourism

The global tourism industry is undergoing some substantial changes. The advance in new technologies and the targeting of a skilled and demanding consumer base means

that hospitality organizations and destinations need new marketing and management tools to meet the expectations of modern tourists and the industry's growing requirements for innovation [24]. Tourism can contribute to this objective by giving intangible water heritage sites a tangible economic value (besides environmental) to help preserve them and any connected water-based ecosystems and biodiversity. Water-based experiences are potential solutions to preserve the environment and the economic value of water and its tangible and intangible heritage [25]. Marina services are part of marine tourism so that they can be viewed in the context of water-based experiences.

Researchers from various countries pay great attention to the development of marinas as one of the region's most promising maritime tourism activities, with a particular focus on environmental protection.

Table 2 ranks the most frequently mentioned trends of marina development. The trends deal with various topics related to strategic communication for marinas, clusters and regional partnerships, studies on environmental conservation and the benefits of marine tourism, the rise of marine tourism, and sustainable tourism planning.

**Table 2.** Trends of sustainable marina development.

| | Salvador et al., 2016. [26] | South Coast Baltic MARRIAGE 2015–2019. [3] | Zacarelli, 2005 [27] | Benevolo, C.; Spinelli, R. 2019 [28] | Martinez Vazquez R.M., Garcia J.M., Valenciano J. de P., 2021 [29] | Zhong H., Zheng Y., Choi D.H., 2020 [30] |
|---|---|---|---|---|---|---|
| Cooperation between regional marinas and public institutions, clustering | x | x | X | x | | x |
| The additional service packages for tourists | x | | X | | x | |
| Marina sustainability as a part of "blue economy" | | | | x | x | x |
| Modern communication system and channels | | x | | x | | x |

All in all, the most notable trends in marina development include ecological and environmentally-friendly infrastructure and the formation of maritime, tourist, and thematic clusters.

It can be concluded that in addition to essential marina services, there is a growing need for additional services that are not directly related to sailing. With a focus on a wider range of visitors, marina managers are constantly focusing on event planning services for both private and public events. Event planning is seen as an additional complementary service with great potential to promote the marina service product [31].

## 4. Research Methodology

The factors and development trends of marina services mentioned in the literature and regional surveys allow us to make assumptions about the expectation of marine service complexes from the consumers. However, in the marinas of the Klaipeda region, a complex study of consumer expectations and analysis of the possibilities of service providers to meet those expectations has not been conducted until now. The purpose of the conducted research was to determine the attractiveness of the marina service complex with the user taken into account.

The marinas of the Klaipeda region are characterized by seasonality. These smaller marinas require a more developed infrastructure to keep ships on shore during the off-season. To attract more users and tourists, they should expand their range of services to potential customers and improve their image.

The aim of the survey was to determine which marina services are most attractive in the Klaipeda region and through which channels information about services provided by the marinas reaches the user.

This study relied on the quantitative survey of visitors to the marinas in the Klaipeda region conducted in May of 2021 to identify the essential components of the marina services. The data was collected by conducting a survey online or interviewing visitors directly. The survey areas included Smiltynės yacht club, Pilies port, Nida port, and Dreverna port.

The total sample size of the survey was 398 respondents, including 239 local and 159 foreign visitors. However, the selected sample size is random and not representative of the population. Random sampling ensures that results from the sample should approximate what would have been obtained if the entire population had been measured [32]. The simplest random sample allows all units in the population to have an equal chance of being selected. The sample size of the research has been calculated using Paniotto's formula, as per Kardelis's study [33]:

$$n = \frac{1}{\left(\Delta^2 + \frac{1}{N}\right)}$$

The survey questionnaire consisted of 14 questions, which were classified as follows:

Questions 1–4 are intended to identify the user: to find out which marina/marinas of the Klaipeda region they visited and for what purposes, from which country they came from and with whom;

Questions 5–9 aim to answer how the respondents learned about the marina and its services and what services and features of the area prompted them to visit a particular marina.

Questions 10–12 are intended to receive feedback from users (customer experience): is the marina easily accessible, were all the advertised services provided, and would you recommend others to use the marina's services?

Questions 13–14 are intended for user recommendations: to express an opinion on what factors would encourage a visit to the marina and what additional services the marina still needs.

The research followed two core ethical principles:

First, this study relied on the principle of goodwill. Respondents were assured that they were not exposed to any risk associated with the research process or results.

The second principle was the principle of justice. Survey respondents were guaranteed confidentiality and anonymity. Responses were anonymized, and the presentation of results contained only depersonalized information.

*Limitations*

This research has potential limitations, such as its limited access to the appropriate type and geographic scope of participants. This survey was conducted in one region of one country, therefore every respondent may genuinely be a random sample. Moreover, conflicts on biased views may be possible—some respondents did not understand questions because of their cultural background or personal view. Another limitation of the survey is the methodological approach. The data analysis was performed using the Excel program and statistical methods were not applied.

## 5. Empirical Research Results and Discussion

The survey aimed to evaluate current marina services and development expectations.

Figure 1 presents the factors of the marina services that enticed customers to come to the marina.

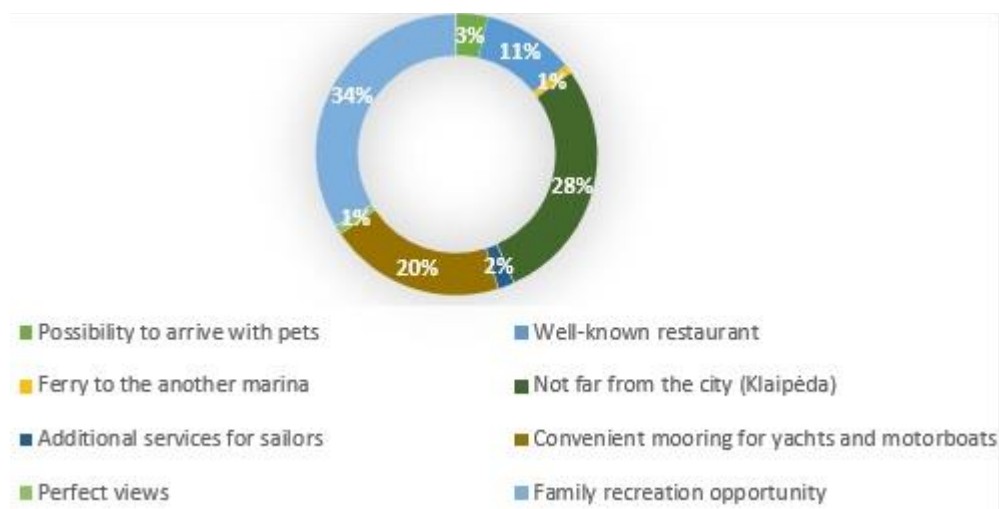

**Figure 1.** The factors of the marina services enticed customers to use the marina.

One of the survey's goals was to identify service products that encourage the use of marina services and what venue aspects were necessary for the service user. The results show that the core factors affecting user experience were opportunities for family recreation (34% of responses), good connections with the central city of the region (28% of responses), and convenient moorings for yachts and motorboats (20% of responses). **According** to the authors' factors presented in the first part, it can be stated that in addition to the primary services of the marina—which are intended for a very narrow category of users, i.e., sailors—there is an increasing focus on a broader range of visitors. The latter are attracted by creating additional infrastructure and service packages for the consumer segment of non-professional yachters.

Table 3 presents the distribution of the marina visitor groups in the Klaipeda Region and their recreational activities.

**Table 3.** Visitor distribution by category and the purpose of visiting marinas.

| User Group/Purpose of Visit | To Spend the Weekend | To Sail | To Take Part in a Personal/Corporate Celebration | To Have a Meal | Other | "To Enjoy, to Have a Walk at the Lagoon." | TOTAL: |
|---|---|---|---|---|---|---|---|
| *Family with children* | 15.8% | 10.0% | 2.5% | 5.0% | | 0.9% | **34.2%** |
| *Company of friends* | 9.2% | 4.2% | | 5.0% | 0.8% | | **19.2%** |
| *Solo traveler* | 2.5% | 2.5% | 0.8% | 0.8% | 3.3% | 2.6% | **12.5%** |
| *A young couple* | 6.7% | 3.3% | 0.0% | 0.8% | 0.8% | | **11.6%** |
| *A mature couple* | 4.2% | 4.2% | 0.8% | 1.7% | | | **10.9%** |
| *Company of colleagues* | | | 9.2% | 0.8% | 0.8% | | **10.8%** |
| *Other (employee)* | | | | | 0.8% | | **0.8%** |
| **TOTAL:** | **38.4%** | **24.2%** | **13.3%** | **14.1%** | **6.5%** | **3.5%** | 100% |

Most respondents visited marinas as a weekend destination (38.4%), followed by yachtsmen (24%). Only 14.1% of respondents came to use the catering services provided by marinas, and 13% used them for personal or corporate celebrations. Nearly a quarter (24.2%) of respondents came to marinas to sail, which means that other visitors used complementary marina services that added value to the marina complex.

Regarding the target visitor groups, families with children were likelier to choose sailing activities than groups of friends. Young couples were more likely to come to rest

or spend the weekend at marinas than to sail. The proportion of holidaymakers and yachters among mature couples was even. Furthermore, groups of colleagues mostly came to participate in events organized by their employer (9.2% of 10.9% of respondents), and only a small number came to dine (0.8%).

The power of modern media communication is closely related to technological progress. With the rapid development of networks and digital high-tech, new media forms emerge endlessly. Therefore, all kinds of media show the trend of multifunctional integration, which requires us to make full use of the latest dissemination of scientific and technological achievements. The results of Figure 2 show that marina visitors mostly found out about the destination or marina services via friend recommendations (37.5%). The most popular e-marketing tools and communication channels were social networks (19.2%) and the Google search engine (15%).

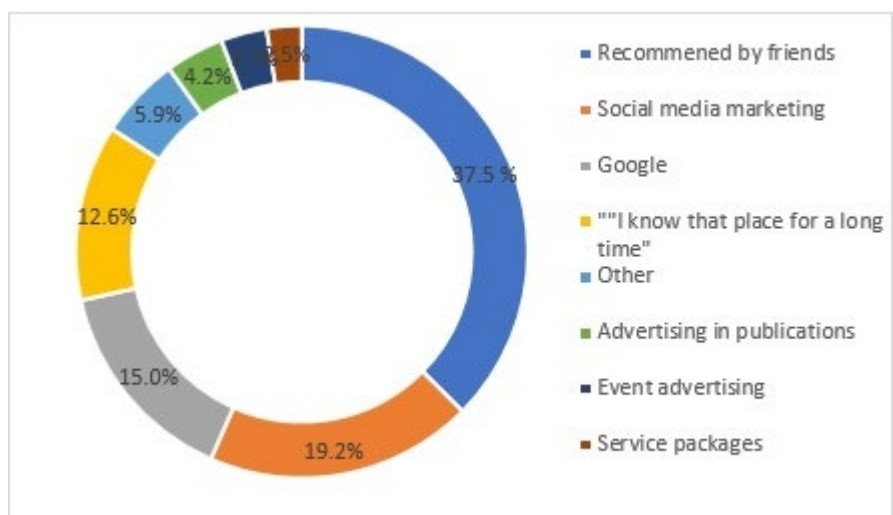

**Figure 2.** Distribution of primary communication channels among users.

Furthermore, respondents were also asked about the accessibility of information on marina services. Over half (58.3%) of respondents indicated that publicly available information was accessible but lacking in detail. Only a third of the respondents believed that the public space provided detailed enough information (30%). In addition, 40% of respondents agreed that available information only covered essential services.

Fewer respondents (15%) indicated that available information was insufficient and misleading. They mainly reported that they had either never heard of marina services or found the information accidentally. Another goal of the survey was to identify service products that encourage marina services and what venue aspects were necessary for the service user. The results show that the core factors affecting the user experience were opportunities for family recreation (31.1% of responses), good connections with other cities (26.67% of responses), and convenient moorings for yachts (20% of responses).

The vast majority (87.5%) of the respondents viewed marinas services positively, with 42.5% recommending local food. A further 20% of respondents indicated good services for yachters, and 13.3% recommended reasonable accommodation. A minority (6.7%) of respondents rated the facilities negatively. Most respondents in this category were not satisfied with the services offered (2.5%), and others (1.7%) did not recommend visiting the marina due to poor infrastructure or were dissatisfied with client service. To enhance the attractiveness of marinas, most respondents (54.17%) suggested using public events for different audiences to attract more users, such as festivals, discos, and live music. They also suggested more open events targeting families with children to ensure a more significant visitor flow and loyalty to the services offered by the marina.

A quarter of respondents indicated that they would gladly participate in loyalty programs for regular marina clients, and 14.2% of the respondents in each group said that

additional marina services should include sports facilities for children and cooperation with sports organizations, art exhibitions, and plain airs.

The client survey results demonstrate that most respondents usually came to the marina for the weekend. Therefore, it can be assumed that they used several services of the marina complex, such as accommodation, catering, and event planning.

This conclusion was justified based on the fact that most respondents were families with children, regardless of the purpose of the visit. The analysis of demand for services proved that the most preferred services would be those targeting families or leisure time. Moreover, customers could be encouraged by loyalty programs for regular clients. They would also benefit from additional information on marina services.

Finally, respondents were asked about their attitudes towards sustainable or green tourism. The majority of respondents (36%) agreed that environmental degradation and sustainability strategies changed public perception of and attitudes towards tourism A quarter of respondents (28%) reported being encouraged to choose more reliable means of travelling.

The most valued elements of marina services were respecting indigenous people and local customs and traditions whilst travelling. Most statements indicated that marina visitors in the Klaipeda region are willing to participate in responsible travel and support sustainable tourism. The analysis of attitudes towards tourism transformation showed that 58% of respondents agreed that today's tourism is inseparable from saving the environment for future generations.

The analysis revealed that a wide range of additional services and entertainment is related to customer satisfaction. Forming the right message and spreading it through the proper marketing channels attracts more users and raises interest in the marina location. Implementing customer segmentation and presenting different value propositions for different segments is crucial. There is a growing trend for marina diversification. In countries with traditional marina culture, a wide range of service products were created in marina complexes, which included accommodation, catering services, and aquatic activities. The results of the research also indicated that the orientation of marinas towards sustainable tourism became a significant expectation of visitors and a value-creating factor when choosing services. It would be appropriate to focus on networking at the regional level and well-targeted marketing communication to increase the attractiveness of marina services to clients.

## 6. Conclusions

From the results presented in this article, it can be concluded that significant changes are observed in the content of marina services moving towards sustainable tourism. The development of marina services has been increasingly oriented toward the main client segment, i.e., yachters and other segments. More diverse visits to marinas also necessitate a better-developed product of marina tourism services. Visitors would like marinas to offer more services: boat storage facilities, catering and accommodation services, safe infrastructure for children, and additional entertainment and public events to make life in marinas more active.

In marinas, and in marina tourism, socially responsible activities and marketing trends are becoming increasingly evident. Environmentally friendly and ecological infrastructure is being created, and the principles of responsible consumption are starting to be applied in marina service activities. Through the application of these measures, many new users from foreign countries who greatly value awareness and sustainability are attracted, and marina users are educated.

In addition, "blue tourism" is proposed for marinas as a conceptual solution that includes maritime, nautical, and marine tourism. For coastal areas that have marinas, its implementation is an element that makes a difference and adds value compared to other locations, which improves tourist offerings and has a decisive role in the economy due to the importance of good territorial, safety, and sustainable environmental planning. For this

reason, "blue tourism" is proposed as a conceptual solution that encompasses maritime, nautical, and marine tourism.

Factors increasing the attractiveness of marinas depend on the choice of the target group. As families with children are one of the most promising segments of marina visitors, this particular group is interested in developing child-oriented infrastructure and sports facilities for children. The popularity of marinas would be enhanced by cooperation with other marinas, sports organizations, the practice of holding art exhibitions and plain airs, the development of restaurant services, and the variety of menus. The research respondents emphasized the importance of information dissemination: increasing communication channels, formulating a relevant message to the target user, and cooperation between marinas, including their recommendation of each other's services.

**Author Contributions:** Conceptualization, A.T.-M.; methodology, D.L.; software, A.T.-M.; validation, A.S., D.L. and A.T.-M.; formal analysis, A.T.-M.; investigation, D.L.; resources, A.S., A.T.-M. and D.L.; data curation, D.L.; writing—original draft preparation, A.T.-M.; writing—review and editing, A.S. and A.T.-M.; visualization, A.T.-M.; supervision, A.S.; project administration, A.S.; All authors have read and agreed to the published version of the manuscript.

**Funding:** This research received no external funding.

**Institutional Review Board Statement:** Not applicable.

**Informed Consent Statement:** Informed consent was obtained from all subjects involved in the study.

**Conflicts of Interest:** The authors declare no conflict of interest.

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
