# Peer review of "Development of Marina Services in the Context of Sustainable Water and Coastal Tourism"

_knowledge, doi:10.3390/knowledge2040037_

Round 1

Reviewer 1 Report

Comments for the Manuscript (Knowledge-1851889)

The paper entitled “Development of Marina Services in the Context of Sustainable Water and Coastal Tourism” examines an interesting subject related to marine and coastal tourism. While I found the overall idea novel and the manuscript good, I believe that the paper needs some revisions before to be published. Below is a list of insights and recommendations that I hope the authors will find useful for their study.

1. Abstract: The authors explain the purpose, methodology and findings of the study briefly in abstract. However, there are simple grammatical and spelling errors in the research. It is recommended that the author(s) correct it with careful reading (line 10, 48 etc.)

2. Introduction: The authors mention about the water and coastal tourism as tourism product and importance of gathering expectations of target market to ensure sustainable marina development. The objectives of the research were also declined at the end of the section. However, they are in discordance with the objectives mentioned in the methodology section. Therefore, the aim and objectives of the study should be revised throughout the text and amended to be consistent. This is the most important issue to be improved by the authors.

3. Literature Review (Theoretical Background): This section was divided into subsection in accordance with the theoretical framework. The authors have reached the basic and foremost studied in the related literature. In this section, it is recommended to argue and/or explain why these objectives were developed, their relevance and importance to indicate the novelty and contribution of the research.

4. Methodology: A quantitative approach has been used in this study. The authors mentioned that the research has an explanatory mixed method design. However, the authors should definitely improve methodology section by explaining universe, sampling, data gathering and analysis procedures that followed.

  • The authors mentioned briefly about the main reason for determining Klaipeda as research universe. They should also mention about the size and structure of universe and explain the reason of quota sampling and how they decide their sample is representative to universe.
  • The main research method was declined as survey. However, there is not any single information about the scales. A comprehensive information about the scales used in the research should be given: (i.e. which scales were applied? How many questions are there in the research scales/ each subscales? Were the scales used as they were in any other research or revised/adopted for this study or newly developed? How was the reliability and validity of construction etc..?).
  • The authors mentioned that explanatory factor analysis (EFA) was applied, there was not any basic information about the EFA results neither as a table nor in the text. KMO, Bartlett’s Test of Sphericity, factor loadings, explained total variance etc. should be given in this section.
  • There was not any information about any other analysis to reach the objectives mentioned in the introduction section.

Since the most problematic sections of the study is the methodology section, the authors should improve this section and provide required methodological information briefly.

5. Research Findings: Authors gave the main descriptive statistics (frequencies) about the participants in this section both in the text and on the tables. It is recommended to give decimals in “total row” as the authors have already applied for “Total column” in Table 1. So the total amount will be 100% (page 8) This section is consistent with the objectives given in the methodology section (but there is not any information regarding to “the changes in the content of the marina services product in transition to sustainable tourism” etc.). The incompatibility problem will be eliminated when the objectives are checked and the unsuitable ones are removed.

 6. Conclusions and Discussions: Conclusions were congruent with the purpose of the study and the findings. However, both the discussions and conclusions sections which were expected to be more interpretative were mainly descriptive and need to be improved. It is recommended that the conclusion and discussions sections not to be written as a re-presentation of the research findings, but by evaluating and interpreting the findings within the scope of the relevant literature.

Author Response

Response 1: The grammar and spelling review has been done.

Response 2: the changes have been made and are highlighted in the paper as red

Response 3: The literature review was supplemented with additional and up-to-date references

Response 4: The factor analysis has not been done, so this phrase has been removed from the methodology. As the methodology part was deeply improved in the article.

Response 5. The results have been recalculated and changed in the table, and the structure of this section has also been changed.

Response 6: Results and discussion parts were combined.

Please, find all article attached, all corrections are highlighted in red color. 

Reviewer 2 Report

The paper addresses an interesting topic but shows several weaknesses that require a complete revision.

Water, coastal, marine, maritime tourism are used as synonyms, while they are not. A vast literature exist on these phenomena that should have been addressed to better identify the scope of the research (which actually is nautical/yachting/boating tourism).

Similarly quite a large number of papers have studied marinas, their offer, their customers and their marketing policy. This literature should be the background for your research. On the contrary, section 2 and especially 3 are quite confused and their contribution is not clear.

In section 4, the contributions available in literature on the sustainability of marinas should have been considered, to provided more focused considerations.

As of the methodology, it is not clear when and where the responses were collected. Furthermore, the explication of aims and objectives is quite confused.

The structure of the questionnaire as well is questionable. For example, the purpose of the visit to the marina includes any kind of responses, of different scope (from “spending the weekend” to “finding shelter for the boat”) and relevance for the study; furthermore, it is not clear if they are mutually exclusive. “Yachtsmen” are considered a group such as “those who came to spend the weekend” but a yachtsman may be in the marina to spend the weekend while sailing. Overall, the structure of the survey is very generic and the analysis of the results is purely descriptive, with no theoretical model guiding the analysis. Finally, the structure of the questionnaire should have been shown in the paper.

Almost absent are the discussion, the theoretical implications and the managerial implications of the study.

 I hope these comments may support the Auhors in improving their manuscript.

Suggested references:

Benevolo C, & Spinelli, R. (2021). Benefit segmentation of pleasure boaters in Mediterranean marinas: A proposal. International Journal of Tourism Research. (23), 134–145. https://doi.org/10.1002/jtr.2403

Jovanovic, T., Dragin, A., Armenski, T., Pavic, D., & Davidovic, N. (2013). What demotivates the tourist? Constraining factors of nautical tourism. Journal of Travel & Tourism Marketing, 30(8), 858-872. https://doi.org/10.1080/10548408.2013.835679.

Lück, M. (2007). Nautical Tourism: Concepts and Issues. Cognizant Communication Corporation, Elmsford, USA.

Lück, M. (Ed.). (2008). The Encyclopaedia of Tourism and Recreation in Marine Environments. Cabi, Wallingford (UK).

Luković, T. (2013). Nautical Tourism. Cabi, Wallingford (UK).

Mikulić, J., Krešić, D., & Kožić, I. (2015). Critical factors of the maritime yachting tourism experience: An impact-asymmetry analysis of principal components. Journal of Travel & Tourism Marketing, 32(sup1), 30-41. https://doi.org/10.1080/10548408.2014.981628.

Raviv, A., Yedidia Tarba, S., & Weber, Y. (2009). Strategic planning for increasing profitability: the case of marina industry. EuroMed Journal of Business, 4(2), 200-214. https://doi.org/10.1108/14502190910976547.

Shen, Y., Kokkranikal, J., Christensen, C. P., & Morrison, A. M. (2021). Perceived importance of and satisfaction with marina attributes in sailing tourism experiences: a Kano model approach. Journal of Outdoor Recreation and Tourism, 35, 100402. https://doi.org/10.1016/j.jort.2021.100402

Author Response

The reviewer‘s suggested references were very useful for the authors, so the sections 2 and 3 were supplemented with the above-mentioned papers:

Benevolo C, & Spinelli, R. (2020), Lück, M. (2007), Lukovic (2013) etc.

In the 2 section marine/ yacht tourism is indicated as a part of water- based tourism. The marina services, which include boat/ yacht services as well as inshore activities, might be indicated as a part of coastal tourism, what was mentioned in the article.

Methodology part was improved. 

According to the comments, “Yachtsmen” are considered a group such as “those who came to spend the weekend”, not only sailing was meant. The aim of coming might be also “participating in the event”, for example a regatta. Moreover, in  to use all provided services: catering, accommodation, shopping ect.

In attached article all corrections are highlighted in red.  

Reviewer 3 Report

Here are some suggestions, and I hope they are helpful for the authors to improve the quality of this manuscript:

1. The cited literature was a bit out-of-date. It would be better if the authors could add some studies published after 2020.

2. "The aim of the research" and "The object of the research" are repetitive in the introduction.

3. Some typos should be corrected, for example, "iwas" in the abstract.

4. The description of the method is not in an academic writing style. Please revise the section.

5. The authors should explain the items on the survey. 

6. Factor analysis was mentioned in the methodology but the results were not presented. 

Good luck with your revision!

Author Response

Response 1: The article has been updated with more recent references: Benevolo,C., Spinelli, C. (2020), Campon- Cerro A.M., Di- Clemente E., Hernandez- Mogollon J.M., Folgado- Fernandez J.A. (2020), Castejón-Porcel, G. (2019)

Response 2: 

The aim of this research study is to identify the main components of marina services that are relevant for visitors in the context of sustainable water and coastal tourism.

It has several specific objectives :

  1. To analyse the changes in the marina services transitioning to sustainable tourism.
  2. To identify ways in which the behaviour of marina visitors shifted towards sustainable tourism.
  3. To determine what factors affect the attractiveness of marina services to consumers.

The primary methods used in this research include literature review of relevant sources and theories and a quantitative survey.

Response 3: Text was reviewed and corrected

Response 5: The research methodology was corrected

Response 6. Factor analysis was not performed in the study, and the statistical dependence of the two indicators was not evaluated. The text has been corrected.

The corrected article has been attached, all corrections are highlighted in red. 

Reviewer 4 Report

The introduction lacks an argument showing clearly and convincingly the importance of the chosen topic, the need to study it and the positioning of the article in the field of knowledge. These tasks require a synthetic review of the literature in support of a compelling argument that your article meets a research need and will make a contribution to the field of knowledge in question.

It is believed that there is a need for greater dialogue with the “theory” of tourism, which will also be necessary in the theoretical review. A specific review of studies dealing with the topic within the area of ​​tourism is suggested.

To what extent does the research problem contain and reflect categories/constructs/variables that come from the tourism-specific literature review itself and how this connects with problem solving of an explicit/implicit agenda of issues in the subarea.

From the discussion carried out, indicate the theory or author defined to support the analysis of the evidence. Include a summary table (replicate the same category table that appears and guides the analysis) at the end of this section, highlighting the main theoretical categories of the study and its theoretical basis.

Methodology: Which categories / constructs / variables were analyzed? Why these variables? Where did they come from/were they taken from? How they connect with theory What are the techniques for collecting, processing and analyzing data? And the limitations of the study?

I recommend this recent paper that can help with some items in your paper: Soares, J.R.R.; Remoaldo, P.; Perinotto, A.R.C.; Gabriel, L.P.M.C.; Lezcano-González, M.E.; Sánchez-Fernández, M.-D. Residents’ Perceptions Regarding the Implementation of a Tourist Tax at the UNESCO World Heritage Site: A Cluster Analysis of Santiago de Compostela (Spain). Land 2022, 11, 189. https://doi.org/10.3390/land11020189

Reorganize and deepen a little the logic of the discussion in the analysis, producing greater tying/systematization between categories, the theory, and the specific theory of tourism, because as it stands, the analysis seems to be based on connections in the form of free association, without leave it tied and explicit, what categories (indicate which categories – topics of discussion – were used to structure the analysis, as they seem to be only implicit), data and analysis techniques were used.

From the discussion carried out, point out the progress/differential of the work in relation to the existing ones on the subject.

Conclusions need to be more robust, supported by the confrontation of the results with the elements of the theoretical foundation section and that make explicit relevant contributions to the field of knowledge, as well as recommendations for future studies. In addition to pointing out the weaknesses of the paper.

Author Response

According to the reviews, the weakest part of the paper was methodology, so it was improved, explained more, than the other parts.
In literature review the information was updated using more or less up-to-date literature. 
In the theory part was decided not explore the topic of marina clusters, but leave this theme for the further researchers, since marina cluster were not deeply analyzed in this survey. 
However, based on the paper you provided, as well as in local cases, it is planned to explore the topic of maritime clusters in the future.
The updated article with all corrections highlighted in red is attached. 

Reviewer 5 Report

No comments.

Author Response

The reviewer has not submitted a single comment. The corrected article is attached. 

Round 2

Reviewer 2 Report

I appreciate the efforts of the authors to improve their paper. However, in my view, most issues I identified are still unsolved.

In particular, the ambiguity between water, coastal, marine & maritime tourism has been only partially addressed.

Similarly, few are the references to literature on marinas management and to marinas sustainability.

The connection between marinas' services and sustainability is not clear, as it is expressed in very generic terms.

As of the methodology, the structure of the questionnaire is unclear and should have been explicitly presented in the paper (possibly in an Appendix). Furthermore, I keep most my perplexities about its construction and the analysis of the results is still purely descriptive.

Still absent are the limitations, the theoretical implications and the managerial implications of the study.

Author Response

  • The part of definitions was extended in the 2nd part of the paper.
  • The structure of the questionnaire has been extended in the part “Research Methodology” as well as limitations. Additional Figures with results were added.

  • Please, find the corrected paper attached - significant changes are marked in red.  

Reviewer 3 Report

Introduction: The literature review is not a type of research method for this manuscript. I suggest the author(s) revise the last sentence of this section.

The author cited the definition from Oxford Dictionaries a few times. However, dictionary definitions are seldom used in academic writing. Citing the definitions proposed by previous researchers would be better.

If the numbers on page 4 stand for the importance ranking, why are the numbers not consecutive? The lists on pages 4 and 5 are a bit confusing. 

Research methodology: How did the author(s) distribute the survey questionnaires? More detailed information should be included. 

Research findings: Some of the findings are in bullet points, which are not formal academic writing. I suggest the author(s) summarize them in paragraphs. 

It seems that only descriptive analysis was used. T-test or ANOVA (some more comprehensive methods) should be used. 

The formats of the citations are not consistent. The author(s) should check the citations and follow the journal author guideline.

This manuscript needs professional editing. For example, "a the targeting of a skilled and demanding consumer base means 258 that hospitality" does not read well (page 6, line 258).

Author Response

Reviewers comment: Introduction: The literature review is not a type of research method for this manuscript. I suggest the author(s) revise the last sentence of this section.

Response: This sentence was revised, as well as the research study aim was described more.

Reviewers comment: The author cited the definition from Oxford Dictionaries a few times. However, dictionary definitions are seldom used in academic writing. Citing the definitions proposed by previous researchers would be better.

Response: Was changed to several researchers definition, with references.

Reviewers comment: If the numbers on page 4 stand for the importance ranking, why are the numbers not consecutive? The lists on pages 4 and 5 are a bit confusing. 

Response: All this part of the paper was generally changed.

Reviewers comment: Research methodology: How did the author(s) distribute the survey questionnaires? More detailed information should be included. 

Response: Corrected, extended: more information about the questionnaire.

Reviewers comment: Research findings: Some of the findings are in bullet points, which are not formal academic writing. I suggest the author(s) summarize them in paragraphs. 

Response: The findings have been supplemented and  summarized in paragraphs.

Reviewers comment: It seems that only descriptive analysis was used. T-test or ANOVA (some more comprehensive methods) should be used. 

Response: One of the limitation of the survey is the methodological approach. The data analysis was performed using Excel program, statistical methods were not applied.

Reviewers comment: The formats of the citations are not consistent. The author(s) should check the citations and follow the journal author guideline.

Response: The citation has been changed in generally according to MDPI paper‘s examples and manual.

Reviewer 4 Report

I still feel that the conclusion needs to move forward and present the weaknesses of the research, in addition to contributions in the field of knowledge and for application in society.

Citations and references are not in the MDPI and journal standards, this is outside the template standards. It seems to me that this unnoticed care testifies against the paper, as it is one of the premises for the paper to be submitted, following the publisher's or journal's template.

I still think I recommend this recent paper that can help with some items in your paper: Soares, J.R.R.; Remoaldo, P.; Perinotto, A.R.C.; Gabriel, L.P.M.C.; Lezcano-González, M.E.; Sánchez-Fernández, M.-D. Residents’ Perceptions Regarding the Implementation of a Tourist Tax at the UNESCO World Heritage Site: A Cluster Analysis of Santiago de Compostela (Spain). Land 2022, 11, 189. https://doi.org/10.3390/land11020189

Author Response

According to Your comments, the citation has been changed in generally according to MDPI paper‘s examples and manual.

The recommended paper was very helpful to improve our paper.

Several parts of the paper has been substantially revised, they are marked in red. 

Round 3

Reviewer 2 Report

The points raised in the previous review have been all addressed. The article is publishable.